# Enhancing Happiness for Nursing Students through Positive Psychology Activities: A Mixed Methods Study

**DOI:** 10.3390/ijerph17249274

**Published:** 2020-12-11

**Authors:** Jeong-Won Han, Kyung Im Kang, Jaewon Joung

**Affiliations:** 1College of Nursing Science, Kyung Hee University, 26, Kyunghee-daero, Dongdaemun-gu, Seoul 02447, Korea; hjw0721@naver.com; 2Department of Nursing, College of Medicine, Dongguk University, 123 Dongdae-ro, Gyeongju-si, Gyeongbuk 38066, Korea; fattokki@gmail.com; 3Department of Nursing, Semyung University, 65 Semyung-ro, Jecheon-si, Chungbuk 27136, Korea

**Keywords:** positive psychology, happiness, nursing students, universities, program development

## Abstract

This study aims to evaluate a program promoting character strengths, positive psychological capital, learning flow, and sense of calling for nursing students. We conducted a concurrent embedded mixed methods study with 51 nursing students randomly classified into an intervention or a control group. The intervention group exhibited significantly higher scores than the control group for positive psychological capital, learning flow, and sense of calling. Program participation experiences were categorized as “change of views about oneself”, “change of views about the world”, “stress relief”, and “practice of positivity”. Among nursing students, this program demonstrated change toward a positive, committed, and meaningful life.

## 1. Introduction

Undergraduates are generally in young adulthood, a time of diverse changes and challenges [1,2], it is a crucial time for individuals to decide on and prepare for a suitable occupation [3,4]. The nursing major guarantees high rates of employment, an important factor for Korean nursing students to consider, as they tend to choose their major based solely on recommendations of parents and others, rather than on their own aptitudes or interests [5,6]. Nursing is therefore not regarded as a vocation but as a practical job choice. After college admission, because of their heavy workload, student nurses have limited opportunities for self-exploration, which hinders future career decisions based on an objective understanding of the “self” and the exploration of appropriate careers and related information [7]. Moreover, nursing students report lower happiness than other undergraduates do because of their course demands, mandatory clinical training in addition to regular coursework, and the burden of the national licensure exam [5,8]. These factors may lead to deterioration of the quality of life of nursing students and are associated with premature turnover among new nurses. High turnover is the most serious problem in nursing in Korea [7], and it adversely affects the provision of stable and quality care for patients. Therefore, specific measures are needed to increase self-exploration opportunities and happiness in nursing students.

Positive psychology is self-exploration that focuses on strengths, rather than weaknesses, and on health and happiness, rather than on illnesses and other problems: this perspective focuses on the best possible outcomes, instead of on previous failures [4]. Seligman, Steen, Park, and Peterson [9] rationalized the concept of happiness using terminology that encompasses a properly defined path to happiness. They argue that authentic happiness comprises the following three elements: having a pleasurable life, defined by experiencing positive emotions to the maximum extent possible, having an engaged life, defined by committing to work that suits one’s strengths, and having a meaningful life, defined by serving and contributing to a greater good, such as family, work, and society [10]. Thus, offering nursing students an opportunity to explore, practice, and cultivate their strengths might make their lives more pleasurable, engaged, and meaningful, thereby increasing their happiness and contributing to securing emotional resources that could aid them in their careers.

Park and Huebner [11] reported that Korean adolescents displayed lower global life satisfaction than their American counterparts. In particular, there was a strong association between school and global life satisfaction among Korean adolescents. Further, Harker and Keltner [12] found that positive emotional expression in college was associated with well-being 30 years later. Therefore, nursing curricula in Korea should include interventions that increase happiness, which encompasses positive emotions and life satisfaction.

In addition to a recent growing interest in positive psychology in Korea, there have been consistent attempts to develop strengths and increase happiness in undergraduates [13,14,15]. However, only three studies considered nursing students [16,17,18] and, in one of those studies, nursing students accounted for only 20% of the total study population [16]. Although the other two studies provided positive psychology-based interventions to nursing students, the programs they used did not involve discovering and utilizing key character strengths.

Thus, this study investigated the effects of a positive psychology-based character strength exploration program that encouraged living a happy life, which encompasses having a positive, engaged, and meaningful life. We anticipate that this program will contribute to developing extracurricular programs that promote happiness and stimulate personal growth in nursing students. We developed three hypotheses for this study:

**Hypothesis** **1** **(H1).**
*The intervention group will show increased positive psychological capital, compared with the control group.*


**Hypothesis** **2** **(H2).**
*The intervention group will show increased learning flow, compared with the control group.*


**Hypothesis** **3** **(H3).**
*The intervention group will show a heightened sense of calling, compared with the control group.*


## 2. Materials and Methods

### 2.1. Study Design

We used a concurrent embedded mixed method design, and qualitative data and quantitative data were concurrently collected and separately analyzed. The results were then integrated into the discussion stage of the study, according to the appropriate research standards [19].

### 2.2. Setting and Participants

Participants were first- and second-year nursing students at a single university, all of whom provided informed consent to participate. Third-year and fourth-year students were excluded due to their clinical training during the semester and the high academic burden of the upcoming national licensure examinations. The sample size was computed using the G*Power 3.1 program. With reference to the effect size of 0.91 for promotion of positivity in a meta-analysis of the effects of positive psychology-based group programs by Eom, Jeon, and Goh [20], we performed the calculations for two-tailed, independent t-tests with an alpha of 0.05, power of 0.90, effect size of 0.91, and two groups, where the minimum required sample size for each group was 20. Thus, considering a 20% attrition rate, we decided to recruit at least 24 participants for each group. Fifty-two students responded to a recruitment advertisement posted on the department’s bulletin board. We assigned them numbers in the order in which their names were listed. The students were then randomly assigned to the intervention group (*n* = 26) or control group (*n* = 26) using a computer program. After excluding one participant who wished to withdraw from the study prior to the beginning of the program for personal reasons, 25 participants in the intervention group and 26 participants in the control group were included in the final study.

### 2.3. Instruments

#### 2.3.1. Positive Psychological Capital

Positive psychological capital was assessed using the Psychological Capital Questionnaire (PPC) [21], in its standardized Korean version (K-PPC) [22]. This 18-item instrument assesses self-efficacy (5 items), optimism (5 items), hope (5 items), and resilience (3 items). Items are rated on a 5-point Likert scale from “strongly disagree” (1) to “strongly agree” (5)—higher scores indicate greater positive psychological capital. Cronbach’s α was 0.89 during questionnaire development, 0.93 in Lim’s study [22], and 0.88 in this study.

#### 2.3.2. Learning Flow

Learning flow refers to the degree of immersion during learning [23], which results in deep retention of knowledge and subsequently experiencing high levels of satisfaction in work and life by nursing students.

Learning flow was measured using the Learning Flow Scale for Adults developed and validated by Kim, Tack, and Lee [23]. This 29-item instrument consists of nine subscales: challenge–skills balance (3 items), clear goals (3 items), detailed feedback (3 items), behavior–perception alignment (3 items), task focus (3 items), sense of control (3 items), loss of self-awareness (3 items), altered sense of time (3 items), and autotelic experience (an activity or a creative work having an end or purpose in itself) (5 items). Items are rated on a 5-point Likert scale ranging from “strongly disagree” (1) to “strongly agree” (5)—higher scores indicate a stronger learning flow. Cronbach’s α for each subscale ranged from 0.65 to 0.90 during development and was 0.92 in this study.

#### 2.3.3. Sense of Calling

Sense of calling was measured using the Korean version of the Calling and Vocational Questionnaire developed by Dik, Eldridge, Steger, and Duffy [24], as validated in Korean by Shim and Yoo [25]. This instrument consists of three subscales—transcendental calling, purpose/meaning, and prosocial motivation—each composed of four items. Items are rated on a 4-point Likert scale ranging from “not at all relevant” (1) to “absolutely relevant” (4)—higher scores indicate a greater sense of calling. Cronbach’s α was 0.85 in Shim and Yoo [25] and 0.82 in this study.

#### 2.3.4. Participation Experience Questionnaire

The participation experience survey was an open-ended questionnaire consisting of the following items:What did you like about this program?What are some of the regrets that you have after participating in this program?What has changed in you or in your environment after participating in the program?

### 2.4. Data Collection and Procedure

This program was structured by modifying the Strength 5 Program (Patent no. C-2013-014687; Strength Garden, Inc. http://www.strengthgarden.co.kr/) using Seligman et al.’s [9] happiness model of positive psychology and character strength. Our program, entitled “Happy caregivers of tomorrow,” is distinguished from existing programs in that it considers nursing students’ characteristics, including their school lives, nursing competencies, and sense of professional calling. Our program also includes videos about the lives of nurses who use their strengths. After restructuring the program, two psychiatric nursing professors and one positive psychology expert assessed its content validity.

The program consisted of two 90 min sessions per week for three weeks (Table 1). Each session opened with “one word of gratitude,” when each participant discussed things they were grateful for. In each session, participants provided examples of using their strengths in their daily lives, after which participants shared their opinions about other people’s strengths. This program was run by the author, who is a positive psychology-based counselor and psychiatric nurse, with help from one assistant facilitator to encourage the students’ active participation. The program was run twice a week outside the participants’ lecture schedules, and participants were divided into small groups by school year to promote heightened engagement. The program took place in a lecture hall where a small group of five to six people could gather.

Data were collected from 7 October to 29 November 2019, after obtaining Institutional Review Board approval (Institution name and approval number blinded for review). Baseline surveys for both groups were conducted prior to the program, and the post-program survey was conducted one week after completion of the six-session program for the intervention group and four weeks after the baseline survey for the control group. To prevent a diffusion effect, the intervention group began after the control group completed the post-program survey. To supplement data that could not be obtained using a quantitative instrument, and to thoroughly understand participants’ experiences of the intervention, a qualitative questionnaire was administered during the post-program survey. To minimize bias, the research assistant collected the data, and was blinded to group allocation. The baseline and post-program questionnaires were given individualized IDs, and collected data were coded to prevent identification of personal information before the assistant entered the data into an Excel sheet.

### 2.5. Data Analysis

Quantitative data were analyzed using IBM^®^ SPSS^®^ Statistics 24.0 software (IBM, Korea Data Solution Inc.). The demographic characteristics of the intervention and control groups were summarized using means and standard deviations or frequencies and percentages. Normality of the dependent variables was tested using Shapiro–Wilk’s test. Homogeneity of the demographic characteristics and dependent variables between the intervention and control group was analyzed using χ^2^-tests, Fisher’s exact test, and Mann–Whitney U-tests. The effects of the program were analyzed using paired t-tests and independent t-tests.

Qualitative data collected through the participation experience questionnaire were analyzed using the content analysis method developed by Elo and Kyngäs [26]. We read the participation experience questionnaire answers repeatedly to grasp their overall meaning, marked words or phrases that contained key thoughts, and clustered similar content into subcategories and named them. Then, the final overarching categories were named.

## 3. Results

### 3.1. Quantitative Results

#### 3.1.1. Baseline Homogeneity Tests of Participants’ Characteristics and Dependent Variables

The mean age of the participants was 19.8 years; among the participants, 38 (74.5%) were women and 13 (25.5%) were men. Twenty-nine (56.8%) had a family income of less than 5 million KRW, while 22 (43.2%) had a family income of 5 million KRW or higher. Twenty-nine (56.8%) had no religion, and twenty-two (43.2%) were religious. There were no significant differences in demographics between the intervention and control groups at the start of the study (Table 2).

At the beginning of the study, there were no significant differences between the intervention and control groups with reference to positive psychological capital (*t* = 1.56, *p* = 0.253), learning flow (*t* = 0.98, *p* = 0.479), and sense of calling (*t* = 1.62, *p* = 0.111). The Shapiro–Wilk’s test confirmed that all variables were normally distributed (Table 3).

#### 3.1.2. Effects of the Character Strength Exploration Program

The means of some variables increased in both groups: positive psychological capital (from 3.39 to 3.86 in the intervention group, and from 3.57 to 3.64 in the control group) and learning flow (from 2.79 to 3.52 in the intervention group, and from 2.94 to 3.07 in the control group). For calling and vocation, the mean for the intervention group increased from 2.85 to 3.04; however, it decreased from 3.05 pre-intervention to 3.00 post-intervention in the control group. The intervention group had a statistically significant increase in positive psychological capital (*t* = 3.59, *p* = 0.001), learning flow (*t* = 4.14, *p* < 0.001), as well as calling and vocation (*t* = 3.15, *p* = 0.003), compared to the control group (Table 4).

### 3.2. Qualitative Results

Students’ experiences of participating in this program were clustered into four categories: “change of views about oneself,” “change of views about the world,” “stress relief,” and “practice of positivity.” To describe each category, we present sub-categories and participants’ direct quotes.

#### 3.2.1. Change of Views about Oneself

Discovering one’s strengths: Motivated by feedback from others, participants were able to discover inner strengths they did not know they had. Their negative self-image changed, and they understood their key strengths.

“I learned about strengths that I was not aware I had, and I felt that I should use them to become a better person.” P7.

“I was not aware of my strengths, but after learning about them through this process, I looked back at my life and was surprised that they were really there.” P22.

Increased self-esteem: The participants focused on themselves as they explored their strengths, and, as a result, began to understand their life values and view themselves positively. Finally, their self-esteem increased, and they became more confident and less self-critical.

“I loved the process of understanding my strengths and recognizing that I am a useful person. Before participating in this process, I had been undergoing inner troubles, but I liked watching myself heal during the sessions.” P9.

“Throughout my life, I could not really focus on myself and find things that I wanted and was good at, but through this activity, I learned to respect myself and I really enjoyed learning about myself.” P25.

#### 3.2.2. Change of Views about the World

Recognizing the values of daily living: Through the expression of gratitude and practice of positivity in each session, participants became interested in minor aspects of their daily lives that they had taken for granted. Through positive perception, they realized the value of their daily lives, and expressed gratitude.

“While writing a gratitude diary, I was able to pay attention to things that I had taken for granted and learned to give them meaning. Moreover, I tried to find positive things even during difficult times.” P10.

“By thinking about the things I was grateful for at the beginning of the program every day, I was able to pay attention to the little things. The moments that I had taken for granted felt so priceless.” P21.

Changes in attitude toward others: As participants shared positive feedback and interacted with their co-participants, they developed greater interest in others and adopted a positive view of them. Further, they expressed gratitude for their families and people around them whom they neglected, and that emotion persisted.

“Sharing each other’s worries and giving advice to one another using our own strengths was impressive.” P14.

“I was able to form a habit of searching for positive qualities in others.” P25.

#### 3.2.3. Stress Relief

Time for rest: Exhausted from the numerous assignments and exam pressure, the participants considered the intervention sessions as a time for physical and mental recovery. The sessions were replete with laughter, with a focus on inner strengths and positivity.

“I was so tired from doing assignments and studying for tests, but every session of the program was so fun and healing. I felt reinvigorated every time.” P11.

“We were so exhausted from the stress of our hectic schedules; this was a much-needed release for us.” P14.

Overcoming negative emotions: Many participants expressed that they had experienced negative emotions, thoughts, and difficulty over the course of their school lives. The intervention moved students’ attention from negative to positive aspects, helping them overcome their problems.

“It was difficult for me to overcome depression, but I think I began to heal quickly.” P9.

“I used to have a negative outlook and when I performed poorly on a mid-term for one class, I kept thinking about it even two months later. This program helped me manage my stress and view the world positively.” P13.

#### 3.2.4. Practicing Positivity

Making positivity a habit: The participants expressed determination to continue expressing gratitude and using their key strengths in daily life after completing the program.

“I practiced developing a strength I was not sure I had; I liked it because I could feel that doing so reinforced my other strengths as well.” P5.

“I became genuinely grateful for even the smallest things. No matter what I did, I could do it gladly. As a result, I was able to treat my friends, family, and other people better.” P6.

Establishing lifelong goals: The participants set out to adopt a positive view and practice positivity, through which they expressed positive anticipation of their future.

“I want to be a strength expert too and help others find their strengths.” P9.

“Before participating in this program, I used to live every day just going through the motions, without commitment. However, while trying to cultivate my strengths, I looked inside myself and promised to myself to take control of my life. As I witnessed my mind change so drastically in a short period of time, I became hopeful that I could become a better person if I carried on with my life with a positive mindset.” P10.

## 4. Discussion

The intervention group showed a significantly greater increase in positive psychological capital than the control group did. This is consistent with Kim and Kim [18], who confirmed improvement in positive thinking among nursing students after a positive psychology-based program, even though the study instrument differed from those used in our study. Our results are also consistent with Bae and Jung [16], who found increased positive experiences and reduced negative experiences and depressive tendencies among participants, including nursing students. Furthermore, these results were supported by our qualitative data regarding the experiences of our participants. Participants’ experiences were categorized into “discovering their own strengths,” “increased self-esteem,” “recognizing the values of daily living,” “changes in the attitudes toward others,” and “overcoming negative emotions.” Positive psychological capital encompasses self-efficacy, hope, optimism, and resilience, it has a positive impact on physical, psychological, and social well-being as well as improvement of academic and work performance [27,28,29]. This program was effective in increasing nursing students’ positive psychological capital and, based on the research above, we anticipate that it will improve their quality of life, satisfaction with their major, and academic performance. Therefore, we recommend incorporating this program in nursing school curricula, especially for mental health nursing, to increase nursing students’ positive psychological capital.

Furthermore, the intervention group exhibited a significantly increased learning flow. These results are consistent with Jo and Park [17]. Moreover, previous results demonstrated a positive correlation between happiness and commitment to one’s academic major and between positive psychological capital and learning flow [8,30]. Similarly, previous studies have found that positive psychological capital lowers academic stress while positively influencing career-related variables, such as academic accomplishment and job involvement [27,28]. These findings are partially supported by our results. Park [8] stated that experiencing major commitment helps students discover a new sense of discipline and induces positive changes, thereby increasing happiness. Further, Seligman [31] presented an engaged life as one of the elements of authentic happiness; therefore, energetically engaging in and committing to the nursing major is important to improve nursing students’ current quality of life and to prepare for the future. The participants increased self-esteem by engaging in activities that bolstered their strengths. According to a previous study, academic stress decreases with increasing self-esteem [32], and reduced academic stress increases learning flow [33]. We infer that our participants’ learning flow increased because of this phenomenon. Additionally, many participants mentioned academic-related stress during the program. Many studies have confirmed stress in nursing students due to schoolwork overload [34,35]. Our qualitative data revealed that participants perceived the program as a “time for rest” and thought the program contributed to their “stress relief.” Positive psychology-based programs such as this could be used in nursing education to help alleviate stress and promote learning flow.

Additionally, the intervention group showed a significantly increased sense of calling compared to the control group. The concept of sense of calling included the purpose and meaning of nursing and prosocial orientation. Similarly, previous reports have shown that positive psychology interventions can provide patients’ lives meaning, positive emotions, and subjective happiness. Moreover, positive psychology-based activity significantly improves participants’ social relationships [14,36,37]. Leading a meaningful life means to utilize one’s inner strengths to achieve something greater than one already has, while failure to find meaning in life despite passionately engaging in life with positivity hinders authentic happiness [31].

The categories that emerged in our qualitative data (“changes in attitudes toward others,” “making positivity a habit,” and “establishing lifelong goals”) represented participants’ determination to continue expressing their gratitude to others, help others, and view life positively, even after the program’s conclusion. These changes confirm that the program helped increase participants’ sense of calling. However, further studies should re-examine the relationship between positivity and sense of calling using instruments that are more objective. Park [8] identified the mediating effect of major commitment on the relationship between sense of calling and happiness in nursing students. An increase in commitment for the benefit of the organization promoted the discovery of new meaning through fostering discipline, which induced positive changes that led to increased happiness. Hence, the effects of our program on learning flow and sense of calling are expected to promote happiness through mutual interaction.

Our participants stated that they felt happy while striving to discover and use their key strengths during the program. According to Seligman [31], using one’s character strengths constitutes authentic happiness. This intervention showed a practical application of Seligman’s ideas through positive psychology. Interest in positive psychology increased in Korea after 2010; thus, Korean research accounts for the majority of the relevant literature cited here, which attests the stress and burnout endemic to Korea’s hectic and complex society.

The major topics of interest in positive psychology include positive mental states, positive traits, and positive institutions, all of which help individuals attain happiness and self-realization [31]. It is critical that current nursing education in Korea establishes resources to help nursing students—who are largely task- and accomplishment-oriented—find and improve their character strengths to succeed in their careers. This study is significant as the first such attempt to integrate a character-strength component into the Korean nursing curriculum.

Nevertheless, this study has the following limitations. First, there is a limit to generalizing only nursing students enrolled in one university to participate. In addition, it is difficult to ascertain the persistence of the post-test measured one week after the end of the program. Therefore, it is necessary to perform repeated studies including expanded subjects in more schools in the future and repeated measurement studies to confirm the persistence of effects.

## 5. Conclusions

This study developed, implemented, and examined the effects of a character-strength exploration program tailored for nursing students in Korea. The program effectively promoted happiness in nursing students, thereby helping them to enjoy, engage in, and pursue meaningful college lives. Use of this program may benefit other nursing students and ultimately positively affect their competencies as healthcare providers. The results provide significant foundational data for use by universities and medical institutions to develop and implement interventions to increase positivity in nursing education. Further research could explore the duration of the benefits of this program among the participants.

## Figures and Tables

**Table 1 ijerph-17-09274-t001:** Character strength exploration program for nursing students.

Session	Theme	Contents
1	Orientation	1. Introduction to positive psychology and character strengths 2. Program orientation 3. Participants’ introduction 4. Oath ceremony
2	Positive views	1. Strength pyramid 2. Master of strengths 3. Strengths identified by others 4. Strength test
3	Remarkable discovery	1. Finding strengths game 2. Drawing a life graph 3. Letter of gratitude 4. Strength seesaw
4	The power of strengths	1. Strength speed game 2. The gift of time 3. Reorganization of daily life 2. Information about the strength-based happiness project
5	Positive relationships	1. Strength coaching game 2. Strength-improving conversation skills 3. Strength Secret Santa
6	Positive realization	1. Strength-based happiness project 2. Strength avatar 3. Postcard of values 4. Strength planner 5. Closing

**Table 2 ijerph-17-09274-t002:** Homogeneity test of general characteristics.

Variables	Categories	Experimental Group (*n* = 25)	Control Group (*n* = 26)	χ^2/^*Z*	*p*
		n (%) or M ± SD	n (%) or M ± SD		
Sex	Female	21 (41.2)	17 (33.3)	2.33	0.199 *
	Male	4 (7.8)	9 (17.7)		
Age (M ± SD = 19.8 ± 1.13)	-	19.7 ±1.15	19.9 ±1.14	−1.162	0.245
Family income	Under 5	14 (27.4)	15 (29.4)	0.02	0.903
(Unit = million won)	Over 5	11 (21.6)	11 (21.6)		
Religion	No	15 (29.4)	14 (27.4)	0.19	0.657
	Yes	10 (19.6)	12 (23.6)		

Abbreviations: M = mean; SD = standard deviation; * Fisher’s exact test.

**Table 3 ijerph-17-09274-t003:** Homogeneity test of outcome variables.

Variables	Experimental Group (*n* = 25)	Control Group. (*n* = 26)	*t*	*p*
	M ± SD	M ± SD		
Psychological capital resource	3.39 ± 0.41	3.56 ± 0.62	−1.56	0.253
Learning flow	2.79 ± 0.42	2.94 ± 0.61	−0.98	0.479
Calling and vocation	2.85 ± 0.40	3.05 ± 0.49	−1.62	0.111

Abbreviations: M = mean; SD = standard deviation.

**Table 4 ijerph-17-09274-t004:** Comparisons of outcome variables between the experimental and control groups.

Variables	Group	Pre-Test		Post-Test		*t*	*p*	Mean Difference (Post-Pre)	*t*	*p*
		M ± SD		M ± SD				M ± SD		
Psychological capital resource	Exp.	3.39	0.41	3.86	0.58	5.61	<0.001	0.47 ± 0.42	3.59	0.001
	Cont.	3.57	0.62	3.64	0.51	1.17	0.252	0.07 ± 0.35		
Learning flow	Exp.	2.79	0.42	3.52	0.60	5.95	<0.001	0.73 ± 0.61	4.14	<0.001
	Cont.	2.94	0.62	3.07	0.58	1.68	0.105	0.13 ± 0.39		
Calling and vocation	Exp.	2.85	0.40	3.04	0.48	4.51	<0.001	0.19 ± 0.21	3.15	0.003
	Cont.	3.05	0.49	3.00	0.41	−0.90	0.376	−0.05 ± 0.34		

Abbreviations: Exp. = experimental group; Cont. = control group; M = mean; SD = standard deviation.

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
