# Peer review of "Enhancing Happiness for Nursing Students through Positive Psychology Activities: A Mixed Methods Study"

_ijerph, 2020, doi:10.3390/ijerph17249274_

Round 1
Reviewer 1 Report
I have enjoyed reviewing your paper. It addresses an interesting and important topic and has applicability beyond Korea. Below I offer a number of comments, for your consideration, that should you accept them will, I believe, strengthen your paper.
- In the first instance I don't believe that you can describe this study as being a mixed methods study. The inclusion of 3 or 4 open ended questions in your follow up questionnaire does not make it a mixed methods design. Rather I think you are best to describe it as a impact /outcome evaluation using a two group pre- post design.
- Please provide a definition of "Learning flow" just as you have for Positive Psychology. Your readers will not be familiar with this term - I had to look it up
- Tables - Please check the t and p values. There are some values that don't look correct to me e.g for age in Table 2 where essentially the age for both groups is identical and in Table 4 where some of the mean difference scores don't seem correct.
- Please edit the text associated with the tables. Avoid repeating statistics in the text. You present these in the table so repeating them in the text is not necessary. Thus just draw the attention of the reader to the important points in the table e.g The intervention group scores increased signifcantly from pre to post test on all three variables while there were no significant increases inthe scores of the control group.
- While analysying mean difference scores is not wrong I do wonder why you didn't use the mean scores in independent t-tests instead - I just think it would have made it easier for the reader.
- Further you either have a significant difference between scores or you don't, please remove references to "a greater significance" in the text in the results section.
- Line 237 - please replace 'interventions" with "intervention sessions"
- In the discussion I think you link your findings to previous research well and have thought about the implications of the findings. However, I would like to see an acknowledgementof the possible limitations of the study. e.g. it would seem to be important that before widely recommending the adoption of the intervention that further research is conducted to determine whether the gains found in this study are enduring. It is well recognised that maintanence of intervention effects is often poor, unless strategies to promote it are put in place. Secondly do you think that the fact that the author of the paper was also the person implementing the intervention might have influenced the written responses to the open ended questions? I think at the very least this possibility should be acknowledged.
Finally overall this paper is well written and has been a pleasure to review.
Author Response
Response to Reviewer 1 Comments
Thank you for this opportunity to revise our manuscript, now titled “Enhancing happiness for nursing students through positive psychology activities: A mixed methods study.” We have carefully considered the reviewers’ comments and provided a point-by-point explanation of how we revised the paper based on the reviewers’ comments and recommendations. We hope that these revisions improve the paper such that you and the reviewers now deem it worthy of publication in International Journal of Environmental Research and Public Health.
I have enjoyed reviewing your paper. It addresses an interesting and important topic and has applicability beyond Korea. Below I offer a number of comments, for your consideration, that should you accept them will, I believe, strengthen your paper.
Point1: In the first instance I don't believe that you can describe this study as being a mixed methods study. The inclusion of 3 or 4 open ended questions in your follow up questionnaire does not make it a mixed methods design. Rather I think you are best to describe it as a impact /outcome evaluation using a two group pre- post design.
Response 1: Thank you for your comment. Although there were only three open-ended questions, the participants provided various opinions in response to these questions. By undertaking a qualitative analysis of the data collected, the authors could gain a deeper understanding of the participants’ experiences in the program and their transformations that cannot be examined by quantitative studies.
The Embedded Design is a mixed methods design in which one data set provides a supportive, secondary role in a study based primarily on the other data type. The premises of this design are that a single data set is insufficient, different questions need to be answered, and each variety of question requires different data types (Creswell, 2007)*. In addition, according to Creswell (2007), qualitative data can be collected not only via open-ended interviews but also through various other forms such as observations, documents, questionnaires, etc.
Based on these considerations, our research can be defined as a mixed methods study because its underlying design, data collection, and analysis were rigorously implemented according to Creswell’s methods.
*Creswell J.W. (2007). Designing and Conducting Mixed Methods Research. Sage Publications: California, US.
Point2: Please provide a definition of "Learning flow" just as you have for Positive Psychology. Your readers will not be familiar with this term - I had to look it up
Response 2: Thank you for this insightful comment. I agree with this suggestion. To assist the reader with ease of understanding, I have added the following definition in lines 104–106: “Learning flow refers to the degree of immersion during learning [23], which results in deep retention of knowledge and subsequently experiencing high levels of satisfaction in work and life by nursing students.”
Point3: Tables - Please check the t and p values. There are some values that don't look correct to me e.g for age in Table 2 where essentially the age for both groups is identical and in Table 4 where some of the mean difference scores don't seem correct.
Response 3: Thank you for your comment. Participants in this study were students in the first and second year of college, and the age ranges between the two groups were considerably similar. However, in the course of this revision, the ages in Table 2 (inserted in line 179; refer to row item for the “Age” variable) were rechecked for normality and corrected according to the Z-values and p-values of the Mann-Whitney U-test utilizing the nonparametric method. Table 4 has been reviewed and corrected for typographical errors.
Points4: Please edit the text associated with the tables. Avoid repeating statistics in the text. You present these in the table so repeating them in the text is not necessary. Thus just draw the attention of the reader to the important points in the table e.g The intervention group scores increased signifcantly from pre to post test on all three variables while there were no significant increases in the scores of the control group.
Response 4: Thank you for your meaningful comments. According to your suggestion, we have revised the content in the results section (lines 189–195)
Points5: While analysying mean difference scores is not wrong I do wonder why you didn't use the mean scores in independent t-tests instead - I just think it would have made it easier for the reader.
Response 5: Thank you for your comment. We agree with your assessment with respect to analyzing mean difference scores. The method used in this study is a simple double difference analysis (Difference in Differences, DID), which is one of the possible methods to analyze the differences between the experimental and control groups. It is also one of the most frequently used analytical methods in the fields of health, psychology, and welfare (Moon, 2012)**. Using the DID analysis, the effect of the program was measured by calculating the differences between the experimental and control groups before and after the program.
**Moon, J. (2012). The impact of public housing on neighborhood land prices (Doctoral dissertation). Pusan National University, Pusan, Korea
Points6: Further you either have a significant difference between scores or you don't, please remove references to "a greater significance" in the text in the results section.
Response 6: Thank you for your comment. Accoridngly, we have removed the reference (“a greater significance”) in the results section (lines 189–195)
Points7: Line 237 - please replace 'interventions" with "intervention sessions"
Response 7: Thank you for your comment. As requested, we have effected this change in line 240.
Points8: In the discussion I think you link your findings to previous research well and have thought about the implications of the findings. However, I would like to see an acknowledgement of the possible limitations of the study. e.g. it would seem to be important that before widely recommending the adoption of the intervention that further research is conducted to determine whether the gains found in this study are enduring. It is well recognised that maintanence of intervention effects is often poor, unless strategies to promote it are put in place.
Response 8: Thank you for your meaningful comments. Per your advice, the limitations of this study were added in lines 335–339 as follows :
Nevertheless, this study has the following limitations. First, there is a limit to generalizing only nursing students enrolled in one university to participate. In addition, it is difficult to ascertain the persistence of the post-test measured one week after the end of the program. Therefore, it is necessary to perform repeated studies including expanded subjects in more schools in the future and repeated measurement studies to confirm the persistence of effects.
Points9: Secondly do you think that the fact that the author of the paper was also the person implementing the intervention might have influenced the written responses to the open ended questions? I think at the very least this possibility should be acknowledged.
Response 9: Thank you for your comments. As stated in lines 153–156, all data in this study were collected by research assistants to exclude the researcher’s bias and minimize the Hawthorne effect. In addition, even if one of the researchers performed an intervention, the analysis and results of the qualitative data were reached by consensus over several meetings with two other researchers.

Reviewer 2 Report
The original article entitled Enhancing happiness for nursing students through positive psychology activities: A mixed-methods study deals with an important matter. The study could have important implications for improving the level of the lifestyle of nursing students in the future.
I appreciated the overall quality of the paper, especially the proper selection of methods, a detailed description of the course of the study and, the criteria for inclusion to the sample.
Comments and suggestions for the Authors
In my opinion the work is prepared very well, but requires a few minor corrections:
- The aim should be clearly stated in the introduction.
- I suggest moving text -line 175-178 - between table number 2 and 3.
- Acknowledgments. This section should include acknowledgments to people or institutions. Maybe would be worth acknowledging to participants of the study.
Author Response
Response to Reviewer 2 Comments
Thank you for this opportunity to revise our manuscript, now titled “Enhancing happiness for nursing students through positive psychology activities: A mixed methods study.” We have carefully considered the reviewers’ comments and provided a point-by-point explanation of how we revised the paper based on the reviewers’ comments and recommendations. We hope that these revisions improve the paper such that you and the reviewers now deem it worthy of publication in International Journal of Environmental Research and Public Health.
The original article entitled Enhancing happiness for nursing students through positive psychology activities: A mixed-methods study deals with an important matter. The study could have important implications for improving the level of the lifestyle of nursing students in the future.
I appreciated the overall quality of the paper, especially the proper selection of methods, a detailed description of the course of the study and, the criteria for inclusion to the sample.
In my opinion the work is prepared very well, but requires a few minor corrections:
Points1: The aim should be clearly stated in the introduction.
Response 1: Thank you for your comments. We believe that the aim of this study is clearly presented in lines 63–67.
Points2: I suggest moving text -line 175-178 - between table number 2 and 3.
Response 2: Thank you for your comment. As suggested, lines 175–178 were moved to lines 182–185, between Tables 2 and 3.
Points3: Acknowledgments. This section should include acknowledgments to people or institutions. Maybe would be worth acknowledging to participants of the study.
Response 3: Thank you for your comments. As per your suggestion, we have thanked the participants in the Acknowledgments section (lines 355–356) as follows: “We express our sincere gratitude to all the participants who were willing and able to partake in the program for this study.”
